# The Problems and Needs of Patients Diagnosed with Cancer and Their Caregivers

**DOI:** 10.3390/ijerph18010087

**Published:** 2020-12-24

**Authors:** Anna Lewandowska, Grzegorz Rudzki, Tomasz Lewandowski, Sławomir Rudzki

**Affiliations:** 1Institute of Healthcare, State School of Technology and Economics in Jaroslaw, 37-500 Jaroslaw, Poland; 2Chair and Department of Endocrinology, Medical University of Lublin, 20-090 Lublin, Poland; grzegorz.rudzki@orange.pl; 3Institute of Technical Engineering, State School of Technology and Economics in Jaroslaw, 37-500 Jaroslaw, Poland; tom_lew@interia.pl; 4Chair and Department of General and Transplant Surgery and Nutritional, Medical University of Lublin, 20-090 Lublin, Poland; slawomir.rudzki@umlub.pl

**Keywords:** cancer, problems, needs, cancer patients, caregivers

## Abstract

*(1) Background*: As the literature analysis shows, cancer patients experience a variety of different needs. Each patient reacts differently to the hardships of the illness. Assessment of needs allows providing more effective support, relevant to every person’s individual experience, and is necessary for setting priorities for resource allocation, for planning and conducting holistic care, i.e., care designed to improve a patient’s quality of life in a significant way. *(2) Patients and Methods*: A population survey was conducted between 2018 and 2020. Cancer patients, as well as their caregivers, received an invitation to take part in the research, so their problems and needs could be assessed. *(3) Results*: The study involved 800 patients, 78% women and 22% men. 66% of the subjects were village residents, while 34%—city residents. The mean age of patients was 62 years, SD = 11.8. The patients received proper treatment within the public healthcare. The surveyed group of caregivers was 88% women and 12% men, 36% village residents and 64% city residents. Subjects were averagely 57 years old, SD 7.8. At the time of diagnosis, the subjects most often felt anxiety, despair, depression, feelings of helplessness (46%, 95% CI: 40–48). During illness and treatment, the subjects most often felt fatigued (79%, 95% CI: 70–80). Analysis of needs showed that 93% (95% CI: 89–97) of patients experienced a certain level of need for help in one or more aspects. *(4) Conclusions*: Patients diagnosed with cancer have a high level of unmet needs, especially in terms of psychological support and medical information. Their caregivers also experience needs and concerns regarding the disease. Caregivers should be made aware of the health consequences of cancer and consider appropriate supportive care for their loved ones.

## 1. Introduction

Cancer has been classified as a lifestyle disease and is a serious problem in modern medicine. Tumours cause around 90,000 deaths annually worldwide, with 120,000 new cases annually. The number of deaths caused by cancer has doubled in women and tripled in men during the last 40 years. The World Health Organization has confirmed that cancer is currently more deadly than cardiovascular diseases [1,2,3]. As reported by the National Cancer Registry in Poland, cancer kills 95.5 thousand people every year. In Poland, cancer is currently a cause of 20% of all deaths, including approx. 40% of deaths of women aged 45–65 years and 30% of deaths of men aged 45–65 years. The mortality due to malignant neoplasms in Poland is higher than the European Union average, by about 20% in men and about 10% in women. WHO predicts this number is going to increase significantly. Within the next 5 years, the amount of new diagnoses every year is predicted to have increased from 19 million and will probably reach 21.7 million by 2030, and 24 million by 2035 [4,5,6].

The dynamics of the increase in the number of cases of malignant neoplasms in Poland is much higher than the dynamics of increase of the population number and is one of the highest in Europe. The number of cases of malignant neoplasms in Poland has more than doubled over the last three decades, in 2010 reaching over 140,500 cases, of which about 70,000 correspond to men and 70,500 to women. The latest data of the National Cancer Registry from 2017 provide 82,450 first-time reports of malignant tumours in men and 82,425 in women. The standardised total incidence rates in 2017 were 565/105 in men and 401/105 in women. It is estimated that for every 100,000 of the Polish population in 2017, 429 people were diagnosed with cancer and approximately 2623 lived with cancer diagnosed in the last 15 years. Today the most common cancer in men is lung cancer, which accounts for about ⅕ of cancer incidence. It is followed by prostate cancer (13%), colorectal cancer (12%) and bladder cancer (7%). Among women, breast cancer is most common, accounting for over ⅕ of cancer cases. It is followed by colorectal cancer (10%), lung cancer (9%), endometrial (7%), ovarian (5%) [6,7]. It has been noticed that in recent years the difference in cancer incidence in Poland, depending on the place of residence (city and rural areas), has blurred. In Poland, rural areas and urban areas are defined according to the administrative criterion. In Poland, localities with the village status have up to 5000 residents. In general, malignant neoplasms pose a greater threat to the lives of men living in rural areas than those living in cities, but due to differences in the age structure, the actual death rate is higher in cities. Only the mortality rate of colon and bladder cancers in men is higher in cities. In the case of women, the mortality rate in rural areas is lower than in urban areas, both for malignant neoplasms in general and for specific cancers, except for gastric cancer [8,9].

Cancer changes the lives of the patient and their loved ones, causing both physical and psychological suffering, as well as negative social and spiritual experiences. Diagnosis of cancer is a threat to one’s sense of security, while feelings and emotions accompanying the disease uproot everyday existence. In addition to coping with the stress caused by the diagnosis, cancer patients have to deal not only with the physical ailments resulting from the illness and its treatment, but also with permanent health impairment, disability, fatigue and pain. Emotional stress and mental problems can cause difficulties in everyday life, such as not being able to work, financial problems and a lack of social support [10,11,12,13,14].

In the course of cancer, three periods have been distinguished. Each of them is accompanied by separate problems and emotions. The diagnosis stage for most patients is tantamount to impending suffering. In this phase, anxiety, anger, fear, denial and despair are the dominating emotions. The chronic stage includes intensive treatment. Remission often occurs and the patient returns to the duties they had before the onset of the disease. However, the treatment process causes the deterioration of well-being, constant fatigue, as well as numerous side effects which significantly reduce the patient’s life quality. The terminal stage can be connected to disappointment due to refusal of further treatment and realization of the inevitably impending death. This period requires particularly careful control of symptoms, ensuring a sense of dignity and improving the quality of life [15].

As the literature analysis shows, cancer patients can experience a variety of needs, as each person reacts individually to the hardships of illness, depending on their personality traits and understanding of their new situation. The passage of time, disease progress and treatment also influence the patient’s mental state. Assessment of needs allows more complex understanding of the experiences of patients and their families, as well as setting priorities for resource allocation, for planning and conducting holistic care [16,17,18]. Medical workers are largely responsible for fulfilling these needs; as shown in the research, the patients’ distress is often caused by lack of the valid source of information and psychological support [12,13].

The influence of cancer on patients’ and their caregiver’s life and needs of patients in the early stages of primary treatment is a very important problem of modern oncology. The research was undertaken because there were few reports of it in the scientific literature. The acquired knowledge will provide a broader view of humans and their needs during cancer, as well as allow the use of appropriate means to fulfil them.

### Objective of the Research

The main aim of the research was to describe the influence of cancer on patient’s and their caregiver’s life and to analyse the problems and needs of patients in the early stages of primary treatment in all areas of life. The acquired knowledge will provide a broader view of humans and their needs during cancer, as well as allow the use of appropriate means to fulfil them.

## 2. Material and Approach

### 2.1. Plan of the Research

The cross-sectional study, based on populations of various areas, was conducted in Clinical Provincial Hospital in Rzeszów, Podkarpackie Oncology Centre between 2018 and 2020. Patients and their carers took part in the survey, so their problems and needs could be assessed in the early stages of treatment. The study included subjects with rather similar characteristics, which was important due to the small size of the sample.

Eligible patients were recruited by the research team (nurse, doctor) in inpatient and outpatient oncology departments, initially on the basis of medical records. Then members of the research team provided these patients with an information pack. After patients consented to the study, the research team obtained from them the data of their formal caregivers. The first contact with the caregiver took place during the patient’s scheduled follow-up visit to the clinic. Caregivers received an identical information pack that the patients had received.

The information package provided by the research group included a letter outlining the objectives and means of the study, a consent-to-contact as well as the non-consent form. If the patient and their caregiver gave their consent, they took part in a 40-min survey at a convenient time indicated separately by the patient and the caregiver (divided into shorter parts in case of the subject’s increased fatigue).

### 2.2. Criteria for the Recruitment

The research group recruited the subjects at the inpatient and outpatient oncology departments. The participants were chosen from adult patients expected to have survived the following 6 months, diagnosed at least 3 months earlier, not suffering from other chronic diseases and being aware of participation in the study of a patient’s family member or significant other who has also served as their caregivers. In order to determine the needs in the early period of primary therapy, participation in the study was offered to the patients during the illness, up to 12 months after the diagnosis. Only patients with tumours were recruited, as patients with blood cancers often do not receive as consistent prognoses as the ones diagnosed with solid cancers. Other excluded groups of patients were these experiencing severe physical pain and psychological burden, minor, not speaking Polish language and those receiving palliative care treatment.

Patients provided information regarding their cancer genetic load and their subjective perception of the first symptoms, they also completed the questionnaire ECOG Scale (fitness scale according to Eastern Cooperative Oncology Group), also called the Zubroda scale, The SDS scale and SAS scale, as well as Visual Analogue Scale. Subjects who underwent surgery were interviewed within one month following discharge.

### 2.3. Representative Group

The research involved 800 patients (78% women and 22% men). 66% of the subjects were village residents, while 34%—city residents. Patients were averagely SD 62 (11.8) years old. Subjects were treated accordingly by the public healthcare system. The surveyed group of caregivers was 88% women and 12% men, 36% village residents and 64% city residents. The average age of the subjects was SD 57 (7.8) years.

### 2.4. Questionnaire

The verification of the interview took place by testing a group of 30 subjects after it was approved by a clinical psychologist. It allowed obtaining epidemiological information, assessing a patient’s condition and needs. The research tool used to test caregivers was a questionnaire allowing to obtain information regarding problem-solving, communication, current role and general functioning.

### 2.5. Method

The ECOG scale (fitness scale according to the Eastern Cooperative Oncology Group, also called the Zubroda scale), was used in the interview. It was used primarily in oncology to assess the patient’s overall fitness. SDS and SAS tests were also used. Both are Likert scales, in which the elements relate to psychological and physiological symptoms using a 4-point scale from 1 (missing, short-term) to 4 (lasting for most of the time or all the time).

### 2.6. Moral Considerations

The Bioethics Committee at the University of Rzeszow approved the study (Resolution No. 2017/12/4). Subjects participated voluntarily, anonymously and were aware of their right to withdraw from the study, the purpose of the study and the time of its completion.

### 2.7. Analysis

The research team used Prism 4.0 software, descriptive statistics and confidence intervals to collect and compare data. Continuous variables are shown using arithmetic means, standard deviations, medians. Statistical characteristics of step and qualitative variables were presented as numerical and percentage distributions with Student t-test or the Mann-Whitney U test. The team assessed internal consistency with Cronbach’s α composite scales and subscales; the unmet needs and quality of life—with linear regression. To determine the correlations, Pearson test and χ^2^ were used for the comparison between the groups. The significance level was *p* < 0.05. The repeatability of answers to individual questions was assessed with Kappa Cohen statistics.

## 3. Results

### 3.1. Patients: Standard Demographics

The subject group included 78% women and 22% men, whose average age was SD 62 (11.8) years. The age of subjects varied from 26 to 85 years. Other descriptive statistics that identify the subjects are included in Table 1.

The data on the most common cancers that the patients suffered from showed that breast cancer in women (38%, 95% CI: 34–42) and colorectal cancer in men (28%, 95% CI: 25–35) were the most frequently indicated occurrences of cancer (Table 2).

Symptoms that occurred in patients before the diagnosis were: weakness (38%, 95% CI: 31–43), weight loss (21%, 95% CI: 19–27), sweating (13%, 95% CI: 8–15), blood in the stool (10%, 95% CI: 8–15), bone pain (7%, 95% CI: 2–12), nausea, vomiting (5%, 95% CI: 2–12), diarrhoea (5%, 95% CI: 2–12). Regular preventive examinations were performed by 43% (95% CI: 40–48) of patients, the remaining 57% (95% CI: 52–60) of patients did not undergo preventive examinations (Figure 1).

### 3.2. Patients: Treatment

Most patients (85%) received chemotherapy. Radiotherapy was used in 23% (95% CI: 13–25) of subjects. Few respondents underwent brachytherapy (10%, 95% CI: 8–17) or surgical treatment (2%, 95% CI: 1–7). Therapy had the least impact on the emotional state of the respondents, as 74% of subjects did not feel its impact on this matter. The respondents who felt that therapy affected their emotional state most often (22%, 95% CI: 19–25) indicated the occurrence of depressive symptoms, sadness, crying and anger. The effect of therapy on the physical condition was noticeable in 39% (95% CI: 31–41) of people, and most often (36%, 95% CI: 31–41) was manifested by the lowered mood. Almost everyone (96%, 95% CI: 88–98) felt that therapy affected their physical condition, which was mostly manifested by fatigue and weakness. The relation between the effects of therapy on mental and emotional state and respondents’ age, gender and place of residence was not confirmed.

### 3.3. Patients: Psychological IMPACT

At the time of diagnosis, the subjects most often felt anxiety, despair, depression, feelings of helplessness (46%, 95% CI: 40–48), disbelief, failure to accept negative information (36%, 95% CI: 24–37), anger, disappointment (13%, 95% CI: 8–17) or indifference (18%, 95% CI: 13–24) (Figure 2 and Figure 3). Most often (76%, 95% CI: 70–80) respondents coped with emotions with family support. 20% (95% CI: 19–30) of people used the services of a psychologist, while few (2%) did not cope with emotion and stress. Most respondents (90%, 95% CI: 88–94) received help from their family or friends. 10% (95% CI: 8–17) of respondents did not receive help. The vast majority (91%, 95% CI: 88–94) expressed a need of support, 89% would like to receive it from their caregiver, 20% (95% CI: 13–24) would like to join support groups, 15% (95% CI: 13–24) would like to have support from medical staff, and 5% (95% CI: 1–7)—from a priest.

Before starting therapy, the respondents most often were afraid of the effects of the treatment (47%, 95% CI: 43–49) and loss of control over their own lives, being dependent on others (39%, 95% CI: 33–42). Patients were least afraid of pain and appearance changes (14%, 95% CI: 11–17). The occurrence of anxiety before starting therapy did not have any significant relation to the respondents’ age, their sex, or their residence. The majority of subjects (76%, 95% CI: 73–78) accepted their appearance resulting from the disease. 20% hid their disease, did not talk about it, avoided contact with people, felt worse than others. Few (4%, 95% CI: 2–7) were ashamed of their appearance, they could not accept it, were isolated and withdrawn. The age of the respondents did not significantly influence their opinions about the impact of appearance changes on their life. It was found that rural residents (82%, 95% CI: 73–88) more often accepted their appearance than urban residents (65%, 95% CI: 61–78). The differences shown were not statistically significant. Over half of the respondents (69%, 95% CI: 61–78) accepted their illness, while 31% (95% CI: 27–38) could not accept the illness and come to terms with the situation, felt overcome by the disease. Attitude towards the disease did not significantly depend on the age of the subjects. Women accepted their disease significantly more often (74%, 95% CI: 70–78) than men (50%, 95% CI: 43–68). Positive attitude towards the illness was slightly more often presented by rural residents (71%, 95% CI: 73–78) than urban residents (65%, 95% CI: 63–68). The differences were not significant for statistics (*p* = 0.5052).

Just over a third (39%) of respondents confirmed that cancer had allowed them to appreciate values that were previously underestimated, most often related to health and life. According to 61% (95% CI: 57–70) of respondents, the disease does not affect the appreciation of previously unnoticed values. 19% (95% CI: 17–28) of the respondents imagined the future believing that it would be good and that they would recover. 81% (95% CI: 77–88) could not imagine the future. Almost all respondents (98%, 95% CI: 92–99) received the necessary support from their family/loved ones. 2% (95% CI: 1–6) of people did not receive such support, claiming that they did not have a loved one. Despite the support, 9% (95% CI: 7–12) felt loneliness, 91% (95% CI: 87–98) did not feel lonely. Men felt lonely significantly more often (23%, 95% CI: 17–28) than women (5%, 95% CI: 1–8). The necessary support from family/caregiver was more often received by women (100%, 95% CI: 97–100) compared to men (91%, 95% CI: 97–99). Slight differences suggested that loneliness was experienced more often by rural residents (94%, 95% CI: 91–99) than urban residents (85%, 95% CI: 78–89). The differences found were not statistically significant. The study attempted to screen for mental disorders among patients. After using the self-evaluation scales, an index of 50 and more, indicating depression, was shown by 31% of subjects. An index of 45 and more, indicating anxiety, was shown by 69% (95% CI: 59–79). Of the entire sample, 21% (95% CI: 19–29) took psychotropic drugs due to diagnosed depression (after the diagnosis of cancer).

Most of the respondents (85%, 95% CI: 79–89) stated that the disease affected their family. 73% (95% CI: 70–80) quit their jobs because of illness. 27% (95% CI: 19–30) of respondents did not have to quit their jobs. 10% (95% CI: 8–17) felt they were worse than healthy people due to their disease and restrictions caused by it. 14% (95% CI: 8–17) felt they were a burden for others, because of the necessity of asking for help or continuous stay in the hospital. The other 86% (95% CI: 79–89) of respondents did not feel they were a burden for others. It was found that people aged 49–60 (88%, 95% CI: 81–89) quit their work significantly more often than respondents aged 61–70 (72%, 95% CI: 70–80) or 71–85 (56%, 95% CI: 51–58). Slight differences also suggested that the impact of the disease on the family was more often mentioned by respondents over 60 years of age. Men more often (27%, 95% CI: 25–32) than women (10%, 95% CI: 8–16) felt they were a burden for others.

The most difficult aspects of the disease were: fatigue (23%, 95% CI: 19–30), fight against the disease (20%, 95% CI: 19–30), fear of the future (20%, 95% CI: 19–30), fear of treatment failure (18%, 95% CI: 15–28) (Figure 4). The most difficult aspects of cancer did not differ significantly among patients in different age groups. Women mentioned fatigue as the most difficult aspect of cancer more often (27%, 95% CI: 18–29) than men (9%, 95% CI: 5–12). Men more often (27%, 95% CI: 19–30) than women (18%, 95% CI: 13–24) mentioned they were afraid of the future. Fatigue was more often mentioned by urban residents (35%, 95% CI: 29–38), while rural residents more often (23%, 95% CI: 19–30) indicated fear of ineffective treatment as the most difficult aspect of cancer. The differences found were not statistically significant.

### 3.4. Patients: Healthcare

75% (95% CI: 70–80) of respondents did not feel properly educated by healthcare professionals about their illness. 25% (95% CI: 19–30) felt fully educated in this respect. Medical journals were the main source of information about the disease (79%, 95% CI: 78–84). Less than half (42%, 95% CI: 37–46) used the Internet, while 28% (95% CI: 19–32) chose other media. Few subjects (18%, 95% CI: 13–24) mentioned the doctor as a source of knowledge about the disease, 10% (95% CI: 8–12) indicated a nurse. The need for information was reported by 86% (95% CI: 81–89) of respondents and everyone (100%) would like to gain knowledge from a professional about treatment (98%, 95% CI: 91–99), course of the disease (78%, 95% CI: 69–79), future prognosis (81%, 95% CI: 79–88), recovery prospects (84%, 95% CI: 81–88). The subject of life expectancy was discussed in 8% (95% CI: 4–9) of patients.

### 3.5. Patients: Symptoms

During illness and treatment, the subjects most often felt fatigued (79%, 95% CI: 70–80) and reported skincare problems (42%, 95% CI: 39–45). The relation between the age of respondents and symptoms they experienced was not confirmed during the study. Slight differences suggested that in the 71–85 age group, pain (7%, 95% CI: 3–9) and behavioural disorders (22%, 95% CI: 19–25) (Table 3).

The analysis of our own research showed that weakness was reported significantly more often by men (45%, 95% of CI: 39–47) than women (22%, 95% CI: 19–30). In the group of men, skin problems also occurred significantly more often (64%, 95% of CI: 62–67) than in the group of women (36%, 95% of CI: 32–68). The place of residence of the respondents did not significantly influence the symptoms they experienced. Slightly more often, weakness was reported by urban residents (38%, 95% of CI: 33–41) than by rural residents (21%, 95% of CI: 19–25).

### 3.6. Patients: Performance Status

Most of the respondents (71%, 95% of CI: 69–75) said that physical symptoms associated with the disease partially limited their basic activities. Total restrictions in this matter were indicated by 9% (95% of CI: 3–12), while 20% (95% of CI: 13–22), of respondents did not feel any restrictions in this respect. Partial restrictions of daily functioning due were more frequently indicated by men (86%, 95% CI: 80–88) than women (67%, 95% CI: 54–69.). The differences were not statistically significant. No relation between the place of residence and restrictions in everyday basic activities due was confirmed (Table 4). The use of the ECOG Karnofsky and Zubrod scale to assess the overall fitness of patients showed that 69% (95% CI: 59–73) of them were unable to work or maintain normal activity, while self-service ability was maintained, 20% (95% CI: 14–26) required temporary care while retaining ability to meet most of their daily needs and 11% (95% CI: 5–16) remained active and experienced minor ailments and symptoms of the disease.

### 3.7. Patients: Needs

Analysis of needs showed that 93% (95% CI: 89–97) of patients experienced a certain level of need for help in one or more aspects. Almost all patients mentioned they had a medium or high need for psychological or emotional support (91%, 95% CI: 89–97), and 86% (95% CI: 81–97) had medium to high needs in the field of medical information. 85% (95% CI: 81–97) of respondents had a medium to high need for financial help. 63% (95% CI: 79–75) of patients needed help to cope with symptoms of the disease an 32% (95% CI: 30–35) required help in everyday functioning. 67% (95% CI: 59–73) of patients had spiritual needs, while 21% (95% CI: 19–30) had social needs. Only 12% (95% CI: 9–17) of patients had high or moderate sexual needs (Table 5). Results of analysis show that the level of needs was statistically significantly related to the patient’s sex. Female subjects more often reported spiritual and emotional needs in comparison to male subjects who reported sexual needs and financial assistance more often in comparison to female patients (*p* = 0.03).

### 3.8. Caregivers

The studied group was 88% women and 12% men. The average age of the subjects was 57 years, SD 7.8. Their age varied from 31 to 85 years. Most of the respondents were married (65%, 95% CI: 61–67), working full-time or part-time (55%, 95% CI: 51–58), with at least secondary education (71%, 95% CI: 68–75). Most caregivers were patient’s family (85%, 95% CI: 81–87), friends (10%, 95% CI: 6–13) and for 5% (95% CI: 2–7) of patients, the caregivers were friends who performed this function informally. Most caregivers did not live with the patients and those who did were mainly patients’ spouses. Caregivers suffered from hypertension (45%, 95% CI: 41–47), insomnia (11%, 95% CI: 6–17), arrhythmias (7%, 95% CI: 1–16), asthma (5%, 95% CI: 1–16), rheumatic disease (4%, 95% CI: 1–16). The vast majority of caregivers had their own family (93%, 95% CI: 91–96), which additionally hindered the level of guardianship due to the excess of duties. Only 10% (95% CI: 1–16) of caregivers partially quit their jobs and they were spouses or children of patients. Other descriptive statistics that identify the subjects are included in Table 6.

The categories of emotional stress reported by the caregivers were: fear (82%, 95% CI: 71–86), sadness (61%, 95% CI: 58–66) and depression (50%, 95% CI: 61–62), anxiety related to the uncertainty of treatment (96%, 95% CI: 91–98). As many as 89% (95% CI: 81–96) of caregivers were concerned about the financial future (Figure 5). The most common source of stress was “emotional distress” (100%) associated with loved one’s cancer. Out of the entire sample, 12% (95% CI: 7–16) took psychotropic drugs due to depression diagnosed after a loved one’s cancer diagnosis.

Almost half of the caregivers (49%, 95% CI: 40–56) felt unprepared for dealing with patient-related problems. Caregivers mentioned their need of better access to healthcare services and resources (78%, 95% CI: 75–80) and of information on the disease itself (76%, 95% CI: 75–80), treatment (89%, 95% CI: 79–90) and prognosis (89%, 95% CI: 79–90). 21% (95% CI: 19–28) of caregivers received sufficient information from healthcare professionals. Others were not satisfied with education. All caregivers sought knowledge independently, giving in order: Internet (86%, 95% CI: 79–90), friends (56%, 95% CI: 49–60), other patients (32%, 95% CI: 30–37), medical journals (24%, 95% CI: 19–30). A very strong positive linear dependence between the source of knowledge and the presented emotional state (+0.993) was confirmed. This means that subjects, who had reliable sources of knowledge, i.e., medical professionals, showed a lower level of anxiety.

Caregivers also reported dysfunction in terms of social roles. As much as 83% (95% CI: 79–90) of caregivers had to take over the previous role of the patient, i.e., engage in the proper functioning of the family, support other family members and provide logistic organization for everyday life. The results indicated worse functioning of the family in terms of problem-solving (71%, 95% CI: 70–73), communication (52%, 95% CI: 49–55), roles in the family (83%, 95% CI: 79–90) and effective involvement (76%, 95% CI: 73–80).

## 4. Discussion

The authors would point out that the discussion was made difficult due to the deficiencies in the professional literature. Cancer is a disease that has significant physical, emotional, social and financial consequences for those affected and their families. In a significant number of cases, the diagnosis of cancer is preceded by a period of gradual, non-specific symptoms or made by routine screening. Many patients are relatively healthy prior to cancer occurrence and, therefore, are not experienced consumers of medical services. After receiving a cancer diagnosis, the patient faces many problems, including fear of death, disfigurement, pain, disability and financial hardships. Typically, the initial response is shock and denial (the duration of which is highly variable), followed by anxiety, depression and inability to function. At the same time, aside from the damage caused by cancer itself, therapy also brings side effects that often lead to significant or permanent health impairment [11,19,20].

The data obtained by NHIS indicates that the likelihood of a patient’s poor health and disability can be increased by cancer at least twice [11]. Fatigue is responsible for a major disruption in patients’ everyday lives, causing physical and mental disorders. This is also confirmed by our own research, during illness and treatment as many as 79% of patients felt weakness, and 23% of patients considered fatigue as the most difficult aspect of the disease. Research shows that fatigue is compounded by pain that is felt by 0.3–0.5 of the patients in active cancer treatment. The impact of physical suffering on functional impairment and several other psychosocial aspects of health has also been documented [21]. In our own research, the majority of respondents (71%) stated that physical pain associated with the disease partially limited their daily basic activities. 9% of people indicated complete restrictions in this respect. Physical impairment and disability can make performing everyday activities impossible. Research by Yabroff et al. shows that adults, who received a diagnosis of cancer, need help in everyday life significantly more often, compared to healthy people [22]. According to the data obtained by the National Health Interview Survey, cancer survivors, who had not experienced any other chronic disease, reported limited ability to perform daily activities two times more often than those who have never experienced cancer or another chronic disease [23].

Aside from physical health problems, there are also psychosocial issues. Emotional stress related to life with cancer diagnosis and treatment, anxiety and stress associated with everyday physical problems can create new mental disorders for cancer patients, or make the already existing disorders worse; this also applies to their families or caregivers. Physical and mental impairment can also make work or performing other social roles impossible for the patient [11]. A comprehensive study conducted in an oncology centre in the US, involving about 4500 patients over 19 years of age, demonstrated a high rate of symptoms of clinical mental disorders [24,25,26,27]. As noticed by Charmaz, Stanton et al. illness often causes psychological burden. Patients may also experience fear of the future, inability to plan, fear of change in sexual functioning, change of their function in the family [28,29]. Analysis of our own research showed that 81% of patients were unable to imagine the future. According to the National Cancer Institute, patients may face problems associated with their faith. Some of them deal with anger, isolation and reduced self-esteem [30,31]. Another important problem affecting the mental well-being of the patient is the feeling of loneliness, which often appears in chronic illness. Our own research shows that only 9% of patients felt lonely and were mostly men. A person can feel alone with the disease even in a large family. Patients are affected by lack of support, understanding, time, constant rush, lack of detailed information about the disease. Difficult relations with the caregiver and family are also a very common problem. They are a source of stress that adversely affects the treatment process [32,33]. Long-term stress, anxiety, low mood, lack of proper support can lead to the development of depression, which is the most common mental illness contributing to the global burden of disability [34,35]. According to our own research, at the time of diagnosis, the respondents most often felt anxiety, despair, sadness and helplessness (40%), while during therapy 69% of the subjects felt anxiety and 21% were treated for depression.

Cancer changes the lives of not only the patient but also their caregiver. It has a huge impact on the family, children, relations between partners. Family members also have psychological needs. Mental problems of family members sometimes are just as severe as the patient’s suffering. Research by Hodges et al. showed that the mental suffering of patients and their informal caregivers were usually simultaneous, although caregivers felt more anxiety than the patient during therapy [36]. Research by Segrin et al. conducted among partners of women suffering from breast cancer has indicated that mental health of the partners positively correlates with stress symptoms in women with breast cancer and that the effects are two-way [37]. This is also confirmed by our own research, as the feelings most frequently reported by caregivers were worries (89%), fear (82%), nervousness (78%), sadness (61%) and depression (50%). Family members and friends of people with cancer often provide significant support by taking over their responsibilities [38]. This requires significant adaptation, which increases stress, often causing adverse health effects, including depression [39]. Research by Clavarino et al. shows worse functioning of the family in terms of problem-solving, communication, and social roles [19]. Very similar results were obtained in our own research, 71% of caregivers mentioned difficulties in solving problems, 52% in communication, 83% in family roles. Most of the changes that occur in a family environment during a disease situation are influenced by the attitude of caregiver towards the disease and the nature of emotional relationships that existed in the family before the diagnosis. The disease situation disrupts the daily rhythm of family life [32,33]. In the studies of Clavarino et al., most common fears of caregivers related to the lack of preparation to deal with patient-related problems, while almost a third of caregivers were worried about the financial effects of the disease [19]. Temporary or permanent interruption of professional activity is very often the main reason for changes in both the structure and functioning of the family, taking over the roles and responsibilities of the patient [32,33]. This can be a source of financial stress, which results from low income, healthcare costs or lack of health insurance. In research by May and Cunningham, as many as 63% of patients mentioned they had problems paying rent, paying off a mortgage, affording everyday spendings as a result of medical debt. Financial needs were also reported as a common problem of oncological patients [40]. The most common reason for the deterioration of the financial situation of a family is interruption or change of employment after the diagnosis of cancer. This is proved by the studies by Spelten et al., where up to 70% of respondents experienced a reduction or interruption in working time or change of workplace after diagnosis or treatment of cancer [41], as well as by the analysis by Hewitt et al., which shows that 17% of subjects were unable to work [23]. In the study group of caregivers, only 10% of caregivers partly quit their jobs and they were spouses or children of patients.

A huge problem among cancer patients and caregivers is not meeting the needs that are most important to them. In our own research, almost half of the caregivers (49%) felt unprepared to deal with issues related to the illness. Caregivers reported a greater need for health care services and resources (78%) and for information on the disease itself (76%), treatment (89%) and prognosis (89%). Almost all patients reported a medium or high need for assistance in psychological or emotional support (91%) and the field of medical information (86%). Sanson-Fisher research shows that for patients and their caregivers these were mainly psychological needs, in particular concerns about the future, but also lack of information, chronic fatigue and inability to do the things they did before [42]. The need for information among patients and their caregivers has also been demonstrated by Kilpatrick et al. According to Rainbird et al., the areas of unmet need among 40% of patients were related to psychological communication and medical information [43,44]. Chapman and Rush, as well as Epstein and Street, also showed that subjects would like to receive more comprehensive information and education [45,46]. It is reported by the members of the American Society of Clinical Oncology, Oncology Nursing Society and the Association of Oncology Social Work that the patients express their need for knowledge resources [47]. According to Boberg et al., patients assess information needs regarding their disease and treatment as very important [48]. This may mean that both patients and caregivers are marginalized in terms of medical discussion about the patient’s treatment and future. Research suggests that medical workers can be unaware that their patients deal with a range of problems [12,13,49,50]. The information should fulfil the expectations and preferences of each patient, as well as be relevant to the individual case [11]. According to Sanson-Fisher, other needs reported by patients are chronic fatigue and the inability to do the things they did before [42], which is confirmed in the studies by Wu and Yao, as well as Rainbird et al., stating that the most common need among patients was help in coping with lack of energy or fatigue (41%) and coping with pain (28%) [44,51].

Our subjects were patients with various types of cancer, therefore the results of the study can be transferred to similar contexts. However, the limitations include e.g., the dependence of the medical system on the Polish economy. The studies concerned the patients in the early period of primary therapy (from 3 to 12 months from the moment of diagnosis). All results, i.e., disease responses, symptoms and needs, were related to this time frame. We would like to add that patients participate in a long-term study, while this research is the first analysis of the conducted studies. The authors hope that the results of the study will be the topic for future publications in the scientific literature.

## 5. Conclusions

Patients diagnosed with cancer have a high level of unmet needs, especially in terms of psychological support and medical information. These needs are priority areas that should be addressed in order to improve the care of cancer patients. Research shows that dissatisfaction with medical information is linked to the development of anxiety and depression. A key challenge for the oncology team is to identify high-risk patients. It is important to distinguish experiencing and transient suffering related to cancer from excessive, disabling suffering requiring psychiatric intervention. Psychosocial research offers the possibility of early intervention and if mental disorders are not detected or treated, they jeopardise the outcomes of cancer therapies, reduce patients’ quality of life and increase healthcare costs. Perhaps one of the better solutions is the widespread inclusion of mental health issues in the training of health professionals in order to meet the needs of oncology patients.

The caregivers also experience needs and concerns regarding the disease. Caregivers should be made aware of the health consequences of cancer and consider appropriate supportive care for their loved ones. Healthcare providers should show sufficient care for patients and their caregivers and intervene at the time of diagnosis rather than in the more advanced stages of cancer.

## Figures and Tables

**Figure 1 ijerph-18-00087-f001:**
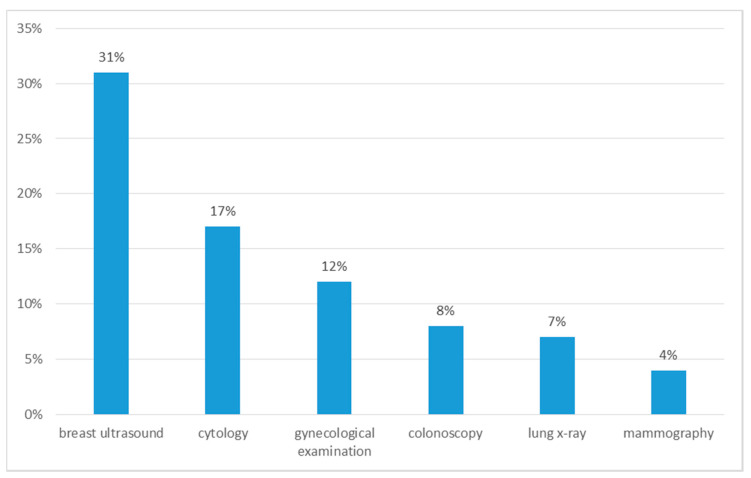
Preventive examinations performed by patients before diagnosis.

**Figure 2 ijerph-18-00087-f002:**
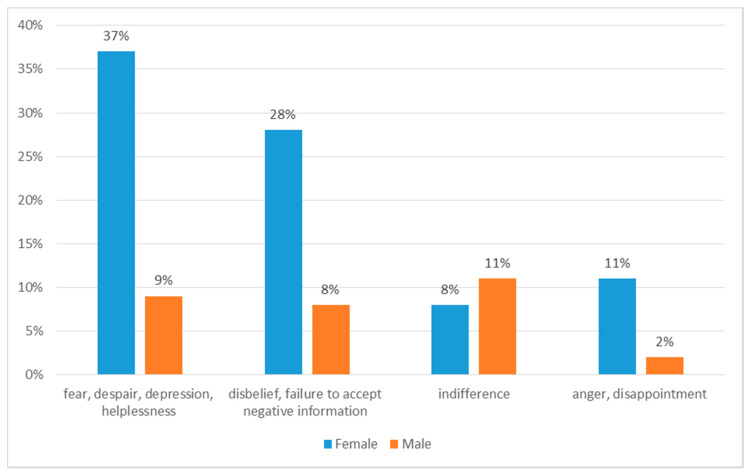
Patients’ responses to the diagnosis of cancer.

**Figure 3 ijerph-18-00087-f003:**
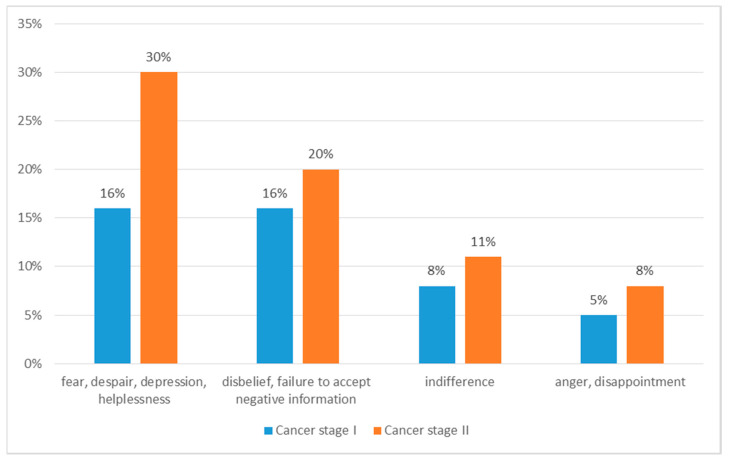
Patients’ responses to the diagnosis of cancer by cancer stage.

**Figure 4 ijerph-18-00087-f004:**
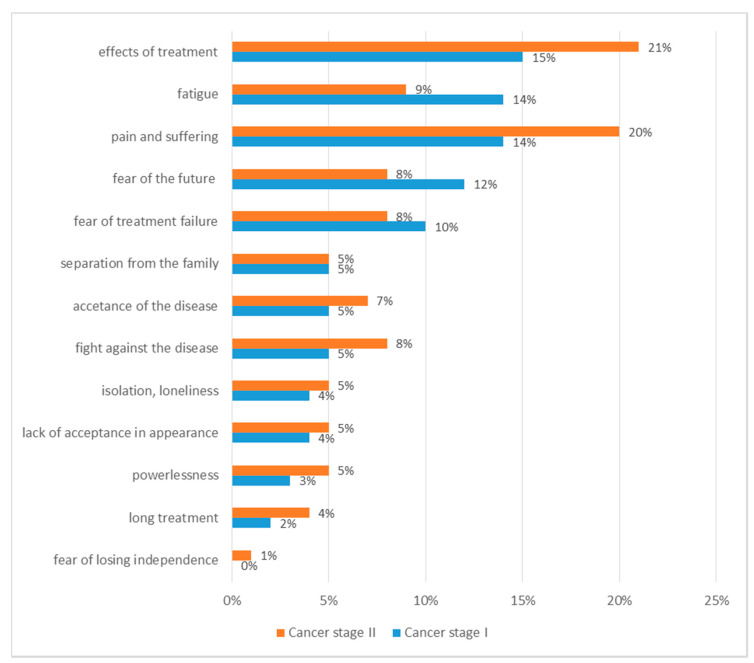
Most difficult aspects of the disease.

**Figure 5 ijerph-18-00087-f005:**
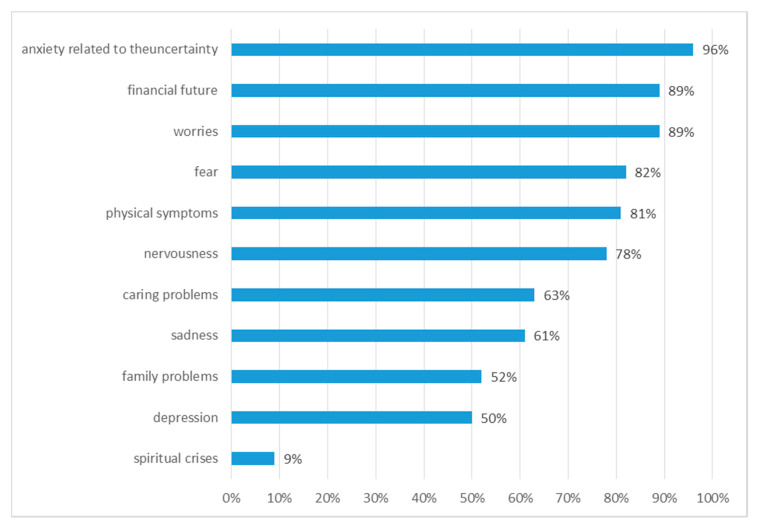
Categories of emotional stress reported by carers.

**Table 1 ijerph-18-00087-t001:** Examined patients—descriptive statistics.

Demographic Information	Total N = 800
Sex	
women	78% (624)
men	22% (176)
Group age	
SD	62 (11.8)
95%CI	<26; 85>
Age of patients	
26–50	33% (264)
51–67	40% (320)
68–85	27% (216)
Place of residence	
city	34% (272)
village	66% (528)
Financial situation	
very good	10% (80)
good	43% (344)
average	35% (280)
bed	12% (96)
Number of children in the family	
1	35% (280)
2	38% (304)
3	27% (216)
Education level	
higher	13% (104)
secondary	37% (296)
vocational	36% (288)
primary	14% (112)
Marital status	
married	85% (680)
widowed	10% (80)
unmarried	5% (40)

**Table 2 ijerph-18-00087-t002:** Incidence of neoplasms among patients.

Type of Cancer	Age			*p*	Sex		*p*	Place of Residence		*p*
	26–50	51–67	68–85		Women	Men		City	Village	
Characteristics % (N)										
Stomach	2% (5)	3% (10)	3% (6)	0.2527	3% (19)	5% (9)	0.5090	3% (8)	1% (5)	0.5090
Nervous system	3% (8)	3% (10)	1% (2)	0.5090	4% (25)	3% (5)	0.1781	3% (8)	3% (16)	0.1781
Lung	22% (58)	18% (58)	18% (39)	0.1781	17% (106)	22% (39)	0.3587	22% (60)	18% (95)	0.3587
Colorectal	15% (39)	15% (48)	23% (50)	0.3587	12% (75)	28% (50)	0.8847	6% (16)	18% (95)	0.8847
Bones	4% (11)	3% (10)	1% (2)	0.5090	5% (31)	7% (12)	0.5440	5% (14)	4% (21)	0.5440
Oral cancer	4% (11)	2% (6)	1% (2)	0.1781	3% (19)	4% (7)	0.5440	3% (8)	1% (5)	0.0909
Breast	25% (66)	25% (80)	30% (65)	0.3587	38% (237)	3% (5)	0.005	30% (82)	34% (181)	0.2527
Ovary	10% (26)	12% (38)	10% (22)	0.8847	12% (75)	0% (0)	0.2527	12% (33)	10% (53)	0.5090
Prostate	5% (13)	14% (45)	10% (22)	0.5440	0% (0)	22% (39)	0.005	10% (27)	8% (42)	0.1781
Lymph nodes	3% (8)	2% (6)	0% (0)	0.2527	2% (12)	2% (3)	0.1781	2% (5)	1% (5)	0.3587
Liver	3% (8)	2% (6)	2% (4)	0.5090	1% (6)	3% (5)	0.3587	3% (8)	1% (5)	0.8847
Other	4% (11)	1% (3)	1% (2)	0.1781	3% (19)	1% (2)	0.8847	1% (3)	1% (5)	0.5440

**Table 3 ijerph-18-00087-t003:** Type of symptoms experienced among patients.

Symptoms	Age			*p*	Sex		*p*	Place of Residence		*p*
	26–50	51–67	68–85		Women	Men		City	Village	
Characteristics/% (N)										
Weakness	30% (79)	30% (96)	18% (39)	0.5090	22% (137)	45% (79)	0.0273	38% (103)	21% (111)	0.2527
Pain	15% (40)	17% (54)	33% (71)	0.1781	81% (505)	73% (128)	0.4134	77% (209)	69% (364)	0.4134
Defecation disorders	3% (8)	15% (48)	20% (43)	0.3587	36% (225)	64% (113)	0.5935	26% (71)	19% (100)	0.0273
Change in skin condition	45% (119)	40% (128)	41% (88)	0.8847	35% (218)	66% (116)	0.0199	41% (111)	38% (201)	0.8847
Weight loss	15% (40)	17% (54)	26% (56)	0.5440	19% (118)	18% (32)	0.9118	21% (57)	19% (100)	0.9118
Behavioral disorders	8% (21)	7% (22)	22% (47)	0.0909	11% (69)	9% (16)	0.7459	12% (33)	7% (37)	0.7459
Other	18% (47)	30% (96)	37% (78)	0.2527	28% (175)	27% (47)	0.9314	24% (65)	17% (90)	0.9118

**Table 4 ijerph-18-00087-t004:** Impact of cancer-related symptoms on limitations in performing simple everyday activities.

Symptoms		Age			*p*	Sex		*p*	Place of Residence		*p*
		26–50	51–67	68–85		Women	Men		City	Village	
Characteristics % (N)											
Effect of cancer on limitations in terms of performing simple everyday activities	Lack of limitations	39% (103)	7% (22)	15% (33)	0.005	24% (150)	5% (9)	0.116	20% (54)	20% (106)	0.9941
	partially limited	48% (127)	87% (278)	74% (160)		67% (418)	86% (151)		71% (193)	71% (375)	
	very limited	13% (34)	6% (20)	11% (25)		9% (56)	9% (16)		9% (25)	9% (47)	

**Table 5 ijerph-18-00087-t005:** Patients’ needs.

Needs	High Needs/Moderate Needs%	95% Cl (%)
Support		
support in dealing with depression	91	87–94
support in dealing with frustration	90	81–93
emotional support from loved ones	89	80–91
psychological support for support groups	20	15–21
psychological support for medical staff	15	10–18
emotional support of the clergyman	5	2–8
Information/Communication		
need information about health status	86	81–90
need information about treatment	98	89–99
the need for education about illness from medical staff	78	67–79
need information about negotiations	81	78–86
need information about health opportunities	84	80–87
Financial		
support in everyday life finances	85	80–87
support in treatment costs	25	17–31
Symptoms/Everyday Life		
help in dealing with symptoms	63	59–67
help in everyday functioning	32	28–38
Spiritual		
changing priorities	39	30–42
help in dealing with the problem of dying	67	61–72
Social		
ability to express feelings	21	18–25
planning the future	19	17–25
support in the functioning of the family	14	11–17
Sexual		
support in intimate life	21	18–25

**Table 6 ijerph-18-00087-t006:** Examined caregivers—descriptive statistics.

Demographic Information	Total N = 800
Sex	
women	88% (704)
men	12% (96)
Group age	
SD	57.51 (7.83)
95%CI	<31; 85>
Place of residence	
city	36% (288)
village	64% (512)
Financial situation	
very good	20% (160)
good	51% (408)
average	20% (160)
bed	9% (72)
Number of children in the family	
1	68% (544)
2	25% (200)
3	7% (56)
Education level	
higher	20% (160)
secondary	71% (568)
vocational	5% (40)
primary	4% (32)
Marital status	
married	65% (520)
widowed	11% (88)
unmarried	24% (192)
Professional activity	
full-time job	45% (360)
part-time job	10% (80)
pension/retirement pension	35% (280)
job resignation	10% (80)
Relationship with the patient	
spouse	60% (480)
child	25% (200)
friend	10% (80)
acquaintance	5% (40)
Living	
living together	65% (520)
separate life	35% (280)

## Data Availability

The data are not publicly available due to privacy and ethical restrictions. The data presented in this study may be available conditionally from the corresponding author.

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
