# Peer review of "The Problems and Needs of Patients Diagnosed with Cancer and Their Caregivers"

_ijerph, 2020, doi:10.3390/ijerph18010087_

Round 1

Reviewer 1 Report

this is an important study about the problems and needs of advanced cancer patients and their caregivers; 

here comes some suggestions:

introduction: it would be good to state more details in the cancer situations in Poland, the demographic, prevalence etc; also, need to define city and village.  

Material and approach:

  • need details on how to approach the patients and their carers ?
  • who gave the questionnaire to them? when will be the questionnaire given ? after seeing the physicians (who may receive some more news about their cancer situations?) or upon arriving the oncology center in the hospital ?      
  • how about the carers ? are they the same return as the patients ?

Result:

  • need more details on the carers, their demographic details 
  • please explain the reason why include occurrence of tumors in the family of patients 
  • patients' response to the diagnosis of cancer ? need to separate with the stage of cancer of patients - 1st, 2nd, terminal,
  • type of symptoms : please correlate with gender, age,    
  • please state the village vs city, how to define village and city in Poland, in terms of population size ? etc, need more details
  • carers needs: please analysis on the kind of cancer their patients are suffering  
  • overall, the results were too lengthy and need to refine and rewrite in a more critical way 

Discussion:

  • the present writing is too lengthy
  • need to rewrite into a more concise and critical way  

Author Response

Dear Reviewer,

We would like to thank the reviewers for their comments. After analysing all the comments, we made the following changes:

  1. In the introduction more details regarding the oncological situation in Poland were added, as well as details of demographics, morbidity, mortality and definitions of a city and a rural area (line 45-80).
  2. The purpose of the research was detailed (line 109-111).
  3. In the “Material and approach” part, the information about the way of recruiting patients and their caregivers was added, as well as the information about the recruitment participants (line 116-137).
  4. In the “Results” part, information about the occurrence of cancer in patients’ families was deleted - we agree that such information was redundant.
  5. In the “Results” part, the information about the patients’ reactions to the diagnosis depending on the stage of cancer was added, however, it should be added that all patients were in the early stage of primary therapy (this information was added in the “Material and approach” part) (line 184-295).
  6. In the “Results” part, the symptoms were correlated with gender, age and place of residence (line 184-295).
  7. In the “Results” part, demographic information about caregivers was added and the types of cancer patients suffered from were analysed (line 338-348).
  8. The description of the results was shortened and changes were made to the “Discussion” part (line 367-477).

I hope that the changes made are satisfactory and this will allow publication. I am asking you to take into account the positive comments of the reviewers that this is an interesting study and a good study.

Sincerely

Authors

Reviewer 2 Report

The authors present a well constructed and well presented discussion of their study and findings.  Relevant literature is cited, a concise introduction is presented, methods are adequately described, significant results are reported. Overall, it is an impressive study and requires only minor language and grammatical review. 

Suggestions:  Various sentences use the wrong pronouns, such as:

line 62: "as literature shows" versus the more common "as the literature shows"

line 77: "...to estimate" versus the more common "describe" as this word better links to the methodology used in this study, cross-sectional, which is descriptive and estimates are not an applicable concept.

line 333: "In own research" vs the more common "In our own research..."

It puzzles me why, after establishing a conceptual timeframe/construct for the cancer pathway (e.g., diagnosis, treatment, after treatment), the authors do not present their findings within this context or more clearly identify the time period along the cancer trajectory they are discussing what findings.  This should be clarified in the paper to help contextualize the identified "needs" and the timeframe along the trajectory these needs were identified.  This is an already clearly established literature with unsurprising findings so presenting these issues in context of time along the continuum might be a more singular way of presenting these findings.  In addition, this division of the cancer trajectory is not "cited" with an author, so it would be helpful if this is a construct BY the authors or they are using this construct developed by other authors.  I find it useful but it needs to be linked to "other research" to help the reader understand the division in the introducton, especially when this division of the trajectory is NOT clarified in the presentation of findings.

Finally, the conclusion is weak.  It reiterates what was found in the study but does not link to other existing research from Europe or other countries as regards similarity or differences, does NOT identify strengths and weaknesses in the design of the study, nor does it identify potential reasons why Poland HC may or may not be similar or different to the rest of Europe.  This weakens the entire paper and misses an opportunity to provide a unique contribution to a crowded literature.

Author Response

Dear Reviewer,

We would like to thank the reviewers for their comments. After analysing all the comments, we made the following changes:

  1. The grammar of the text was reviewed.
  2. The studies concerned patients in the early period of primary therapy, in the period of the disease from 3 to 12 months after the diagnosis. All results, i.e reaction, symptoms and needs were related to this time frame (line 111-137). We would like to add that patients are participating in a long-term study and this research is the first analysis of the conducted studies. Another assessment of problems and needs in the same group will be carried out 12 months to 2 years after the diagnosis and from 2 to 3 years after the diagnosis. This limitation was presented in the discussion (line 470-477).
  3. Information regarding the research group was detailed, the mistake in the topic was corrected.
  4. The requests were corrected, as suggested (line 479-501).

I hope that the changes made are satisfactory and this will allow publication. I am asking you to take into account the positive comments of the reviewers that this is an interesting study and a good study.

Sincerely

Authors

Round 2

Reviewer 1 Report

an important study about problems and needs of patients diagnosed with cancer & their caregivers.

some suggestions:

  1. the introduction needs to be strengthen, and highlight the problem / gap in literature
  2.  method: adequate
  3. results: suggest to revise, and can combined the tables/figures to streamline the results; and avoiding using long paragraphs 
  4.  discussion: seems largely uncritical, please relate your findings with literature, and reorganize the major and important findings      

Author Response

Dear Reviewer,
Thank you for important comments on my manuscript.
I made corrections in the manuscript: I corrected the conclusion of the abstract and shortened the introduction and conclusion of the manuscript.We pointed out that the discussion was made difficult due to the deficiencies in the professional literature.

Kind regards,
Authors